# A Strategic Interpretation of Landscape through Interaction between Natural, Built and Virtual Environments: The Case Study of Piazzola sul Brenta

**Greta Montanari [1,\*], Andrea Giordano [1], Gianmario Guidarelli [1], Federica Maietti [2]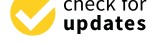 and Elena Svalduz [3]**

[1] Department of Civil, Construction and Environmental Engineering, University of Padua, 35131 Padova, Italy; andrea.giordano@unipd.it (A.G.); gianmario.guidarelli@unipd.it (G.G.)

[2] Department of Architecture, University of Ferrara, 44121 Ferrara, Italy; federica.maietti@unife.it

[3] Department of Cultural Heritage, University of Padua, 35131 Padova, Italy; elena.svalduz@unipd.it

[\*] Correspondence: greta.montanari@phd.unipd.it; Tel.: +39-3497004342

**Abstract:** The need to digitize data as an analysis tool is increasingly a topical issue, also because it is a tool of common interest for several disciplines. This new research merges with the iNEST project (Interconnected Nord-Est Innovation Ecosystem), referring in particular to Spoke 4: city, architecture and sustainable design, which aims at extending the beneficial effects of digitalization to the areas of "Nord-Est" Italy. The project started in August 2022 and will end in August 2025. Knowing that the trend of urbanization, metropolis living, and climate change is related to psychologically stressful situations as a result of environmental stressors, this research aims to analyze if living in a place surrounded by natural and valuable artificial elements (i.e., historical architecture, art) is crucial in generating health and psychophysical well-being. This paper presents the case study of Piazzola sul Brenta, a small town in the Veneto region on which a territorial analysis was carried out to understand the existing dynamics between the natural and artificial environment, using the literature and historical maps. Since this research began recently, with only preliminary and partial results so far, this paper focuses on the interdisciplinary discussion developed around this topic, showing the first part of a research that aims to create models of urban and landscape contexts that enable in-depth analysis and the prefiguration of strategies for regeneration.

**Keywords:** survey; modeling; architectural history; landscape; representation; visualization; restorativeness; sustainability; historical heritage

## 1. Introduction

Using survey and modeling as a tool to analyze the territory and anticipate possible risks is becoming an increasingly topical theme. The assessment of threats related to environmental issues, including climate change, and the conceptualization of solutions in risk reduction, post-disaster participatory actions, governance strategies, monitoring and tools for an aware design applied to the conservation of the historic built heritage [1] are more and more central topics. In fact, the strategic importance of having the landscape modeled is a necessity both from the point of view of conservation and the mitigation of future events and risk management is a global issue, especially in the AEC (Architecture, Engineering and Construction) field. The failure to adequately manage risks may not only lead to difficulties in meeting conservation and enhancement objectives but also influence land-use planning and urban spatial design in the future growth of cities.

Documentation as a tool for knowledge is the first step in creating models of urban and landscape contexts that enable in-depth analyses and the prefiguration of strategies for regeneration. In addition to traditional methods of representation and description at a territorial scale (planimetric representations and cartography in general), today, the

development of 3D surveying technologies and 3D modelling systems enables new approaches in the knowledge and representation of the territory. Digital representations and city models make it possible to include several data simultaneously (urban, territorial, social, cultural) [2] (pp. 2165–2182).

The topic of landscape dynamics is broad and wide, including and intersecting different disciplines and different applications in surveying and modeling coming from digital technologies [3] (pp. 23–32). In addition to 3D technologies for data capturing at landscape and territorial scale, the use of historical maps is essential in analyzing changes over times and assessing built and natural environment, also from a spatial-temporal point of view [4]. For this reason, the first part of the research here outlined focuses on the historical maps available, compared with the state of the art related to the case study analyzed. This method of studying the landscape is a valuable support to understand the past dynamics between the natural and artificial environment, but it also helps to understand how human actions influenced the landscape and how it could be influenced in the future in a positive way, by addressing mistakes or issues resulted from past experiences.

Since the scientific community concurs on the regenerative value of greenery and natural spaces, recognizing that the connection with natural environments indeed encourages the use of involuntary attention and can have restorative benefits on people's affect (lowering stress) and cognition (i.e., memory and attention) [5,6], the aim of this research is instead to analyze what benefit the citizen has from living in a place where there is a mixture of natural and historical valuable architecture.

A place with its own historical identity certainly has an impact on inhabitants and, as Steadman underlined [7], there is an importance of the physical features and conditions in the construction of place and place meanings based on its environmental attributes. In his paper "Is It Really Just a Social Construction? The contribution of the physical environment to sense of place" he also notes that physical characteristics influence the symbolic meanings of the landscape. As a result, the loss of place physical character and identity would influence people's perception and fondness for places [8], highlighting the deep connection between human psychological sense of place and the physical space.

The awareness of being in an historical place, with a memory and a meaning of its own, could be a factor with an influence on psychophysical well-being? To study this topic, it seems essential nowadays to use technologies allowing accurate three-dimensional modelling combined with the possibility of aggregating information and features of different kinds, such as BIM (Building Information Modeling) environments, to support the project development.

Moreover, due to the rapid development and adoption of BIM and BIM-related digital technologies, the use of these tools for risk management has become a growing research trend leading to a demand for a thorough review of the state of the art of these developments [9] (pp. 88–98). Recent research had shown that BIM could not only be used as a support for project development process as a systematic risk management tool [10], but it could also serve as a core data generator and platform to allow other BIM-based tools to perform further risk analysis. For example, thinking about CIM (City Information Modeling), which helps in the search for information on future demands, providing a holistic view of the city, it is clear how useful it is in the urban context to analyze and digitalize data for strategic future planning. In fact, the growing demand of the population has caused serious problems for cities and has become one of the main challenges for city managers [11]. In the urban context, a systematic control of data related to the city dynamics could help designers and urban planners to solve the problems of traffic congestion, accessibility, and the potential impact of natural disasters. Regarding this latter point, the need to digitize data becomes even more urgent if we think of the recent disasters caused by climate change. The possibility of programming simulations of possible cases, based on collected data or imported data, makes new technologies a fundamental tool for strategic design and mitigation. Furthermore, referring to the representation of the landscape, we can see new technologies as a tool for overcoming the gap in the scale representation of a

subject which, otherwise, is significantly affected by the limit dictated by the finite nature of paper supports [12] (pp. 999–1019).

At last, no less important is the possibility that BIM can give from a historical point of view, providing a database, giving the possibility of having a digital heritage as computer-based materials of enduring value that should be kept for future generations [13]. In fact, digital tools enable the general public to take advantage of the easy understanding of the historical contents displayed, graphically summarizes data acquired from archive and on the field, introducing a new use and understanding of cultural heritage [14] (pp. 813–820).

## 2. Research Framework

This research merges with the iNEST project (Interconnected Nord-Est Innovation Ecosystem), financially supported in the frame of PNRR Program (the Italian National Recovery and Resilience Plan). iNEST aims at extending the beneficial effects of digitalization to the key specialization areas of "Nord-Est" Italy (Friuli-Venezia Giulia, Veneto and the Autonomous Province of Trento e Bolzano): industrial and manufacturing, agriculture, marine and mountain environment, architecture and construction, tourism, culture, wellness and food are the fields addressed. The iNEST Research and Innovation Program is organized according to a structure composed of one Hub (University of Padua) and nine Spokes, involving all the Universities of the North-East as well as the main Research and Technology Transfer Institutions active in the area. It is formally a consortium founded by eleven research institutions, nine of which are universities, which in turn identify the nine Spokes, each responsible for a specific research line. The interrelationship between individual lines and research institutions is guaranteed by a complex system of affiliations. The Hub is the body responsible for obtaining the final objectives of the project, each individual Spoke is responsible for carrying out the various research activities. In particular, this research refers to the area of Spoke 4, which concentrate on the theme of "City, Architecture and Sustainable Design".

Given the breadth of topic of Spoke 4, the initial difficulty of the research was to choose a specific theme to deepen and to coordinate with more exponents/researchers from different disciplines. In fact, the trend of urbanization, metropolis living, and climate change is related to psychologically stressful situations as a result of environmental stressors, which includes noise, crowding, traffic, and pollution [15] (pp. 451–464), [16] (pp. 1–11). Improving the quality of life for the ones stricken by those stressors, which include the elderly, migrants, and those with disabilities, calls for proof from numerous fields, which includes psychology, architecture, design, and information technology [17] (pp. 1936–1951).

Based on these assumptions, this research adopts an interdisciplinary method to analyze the interplay among humans and their surroundings and examine the impact of nature and constructed environments on emotions, behaviors, lifestyles, and health. Bringing several disciplines together in a single research therefore becomes crucial. The aim is not only to analyze the space from a physical point of view, but also to understand the reaction that this space triggers in the visitors and tourists and the dimension of involvement with a place that can arouse in the citizen and local communities. Place attachment concept is closely linked to the affective aspects of environmental meaning [18] and as Hidalgo and Hernandez write it refers to *"the development of an affective bond or link between people or individuals and specific places"* [19]. This is a concept expressed through the interplay of affects and emotions, knowledge and beliefs, behaviors and actions [20] and it develops when *"a place is well-identified and felt significant by the users and able to provide condition to fulfil their functional needs and supports their behavioral goals better than a known alternative"* [21]. Recent research locates place attachment within the psychological (emotion and feeling) and the functional (dependence) domain of environmental experience [8] and study whether and how this sense of belonging to a place is manifested, could be the key to a conscious and careful regeneration. As described in the research of Ujang and Zakariya: *"In the context of place regeneration, the sense of belonging, degree of attraction, frequency of visits and level of familiarity are indicators of place attachment"* and these factors are therefore

fundamental source of knowledge to understand a place and the relationship that the individual has with it, revealing the need "*to consider the users' feelings and reactions towards the attributes and characteristics of an urban place*" [8].

The scientific community concurs that natural environments can play an important role in generating health and psychophysical well-being; however, political strategies regularly undervalue the poor effect of nature deprivation, especially on children, older people, and minorities. Being in contact with nature certainly encourages the usage of involuntary interest and might have restorative benefits on people's affect (reducing stress) and cognition (i.e., memory and attention) [5,6].

Concerning this specific topic, the space of the garden is in itself historically a place of *otium*, where psychophysical benefits had been noticed and used to positively stimulate the citizen: as an example, the garden in which Plato organizes the Philosophy Academy: "Having returned to Athens, he lived in the Academy, which is a gymnasium outside the walls, in a grove named after a certain hero, Hecademus, as is stated by Eupolis in his play entitled Shirkers: In the shady walks of the divine Hecademus. Moreover, there are verses of Timon which refer to Plato: Amongst all of them, Plato was the leader, a big fish, but a sweet-voiced speaker, musical in prose as the cicala who, perched on the trees of Hecademus, pours forth a strain as delicate as a lily." [22].

As case studies the research group defined two historically important villages which have relevant characteristics for the topic: Piazzola sul Brenta, in the Veneto region, and Aquileia, situated in the Friuli-Venezia Giulia region.

The research carried out so far is concentrating on the first of the two case studies, Piazzola sul Brenta, which had been chosen for multiple reason, the most important of which are its valuable architecture around which the village historically revolves (Villa Camerini-Contarini), green places integrated with the built space, stretches of water. All of these are regenerative aspects that the research team intends to explore, to understand how natural and artificial spaces interacts with each other, in a small-town context, and how this relation benefits the life quality of citizens.

Therefore, the research to be carried out will go not so much to deepen the regenerative value of greenery and natural spaces, as this is a widely studied topic, but the regenerative value that the union of the natural element and the valuable artificial element can have (historical architecture, artworks).

In fact, despite the developing frame of proof at the mental advantages of nature, much less is known in regard of the restorative impact of historical places and further research is needed. For example, it is still unclear whether historical environments can have positive impacts in terms of well-being, but also cognitive functioning, and emotional state (subjective or physiological). Historically, there are countless examples in which the mixture of architecture and natural environment (green and blue) manages to create a regenerative space [23] (pp. 31–45), [24] (pp. 165–169); [25]; [26] (pp. 100–103); [27]; [28] (pp. 417–437); [29]; [30] (pp. 293–297); [31] (p. 46).

Initially conceived as a place devoted to *otium*, well represented by Petrarch's fourteenth-century home in Arquà, the Veneto villa in the Renaissance, as an integrated complex of open spaces, gardens, "broli", and buildings offers us an opportunity to think about these issues, even though examples closer to the context of Piazzola sul Brenta [32] (pp. 452–460), [33] (pp. 443–452), [34,35]. The combination of humanistic *otium*, inspired by Platonic experience, and agricultural functions is extensively described in the Four Books by Andrea Palladio, who was its main interpreter and the designer of Villa Contarini at Piazzola. Another important factor in a positive human-environment interaction is residential satisfaction and quality.

Nowadays, this aspect has also become central, as a result of COVID-19 confinement in houses, which has brought greater interest about building, neighborhood, cities and province quality [29,36–38].

Moreover, the demographic challenges of an increasingly older population require rethinking the quality of residential contexts, providing environmental and social support

to avoid social isolation [39] (pp. 224–232). Several factors have been individualized as predictors of resident satisfaction (e.g., design characteristics, community engagement); however, the relationship between restorativeness and resident satisfaction has been little investigated [40–42].

This research aims to testify also how new technologies have the ability to drastically change the approach study in those field and how they can be used in a strategic and interdisciplinary way. The current need to interpret and represent the knowledge of a city and the dynamics between natural and built historical spaces, lead us to combine digital and traditional research methods, developing a strategic visualization of those spaces.

Regarding this topic, the current research field presents some interesting studies that will be considered. For example, the initiative *Visualizing Venice/Visualizing Cities*, founded in 2010, presents an analysis of art, architectural, and urban history in the urban space, underlining the importance of *"how new technologies have the capacity to "revolutionize" research and teaching by implementing collaborative theory and practice in the field of Digital Humanities."* [43] (pp. 171–187).

In this specific field, where space is not only physical space but also the perception that the person crossing it has, the interdisciplinary approach is of fundamental importance. In fact, the data collection methodology and their interpretation cannot omit the integration of multiple devices. Therefore, we support a methodology to interpretate datas, *"that supports an operational verification of the meaning of places, which analytically focuses on emotions and meanings and can be used as a strategy for studying not only the territory and the environment, but also the landscape, understood as the result of the perception process"* [44] (pp. 135–142).

## 3. An Interaction between Natural, Built and Virtual Environments

Within the iNEST project, a focus on sustainability was developed, combining the topics of social, environmental and economic sustainability in a single project. This research is framed within Spoke 4—*City, Architecture and Sustainable Design*, in particular referring to the Research Topic (RT) 3.1—*Interaction between environments and human beings in the construction and sustainable design sectors* [45].

Collaborating in this project, there are eleven founding members, nine universities and two institutions: University of Padua, University of Verona, Ca' Foscari University of Venice, IUAV of Venice, University of Trento, University of Bolzano, University of Udine, University of Trieste, SISSA of Trieste; by the National Research Council (CNR) and by the National Institute of Oceanography and Experimental Geophysics OGS.

Therefore, this newly born research is an interdisciplinary and knowledge transfer activity dedicated to social innovation supporting the resilience enhancement and it focuses on the interaction between natural/built/virtual environments and the way in which humans think and act and how places shape us as individuals and communities. It also aims to invest its scientific outcome in informing the design and development of new socially sustainable, inclusive, accessible spaces and interior design products.

Within this context, the aim is to exploit the interdisciplinarity characteristic of this project and to investigate in depth, from both a historical and physical spatial point of view, the built elements of greater importance within the two chosen case studies: Piazzola sul Brenta e Aquileia. Both of them are located in the North-East Italy, a fragile and unique territory, in which the necessary care and maintenance of an articulated landscape is compared to the ethical commitment of design, a widespread historical architectural heritage within a territory characterized by significant environmental risks. In particular in Veneto the region, we are faced with a landscape characterized mostly as a reclamation landscape, which actually has the specific feature that defines a "cultural landscape"; in the *Manifesto per la tutela e la valorizzazione del paesaggio della bonifica del Veneto Orientale*, it is possible to understand how *"the reclamation landscape is an outcome of a long time work devoted to change the water system existing between lagoons and rivers, so regenerating a land to be employed for agriculture, human settlements, industry, tourism."* [46,47]. The reclamation landscape is a key element that characterizes the geographical and environmental infrastructure of

eastern Veneto, together with the basin of the river Piave, near the area of Cadore, the Piedmont system of the hills of Treviso and valleys from land reclamation, right next to the Venetian lagoon [46]. Additionally, the river Brenta, an important natural element for the chosen case study, is part of this landscape and is immersed in what R. Salerno calls a "new landscape", "*almost completely artificial, marked since it appeared, by a series of land and water infrastructure and very specific buildings (canals, banks, roads between small farms, turning bridges, basins, water pumps) which permitted and still today safeguard both a stable emergence and a civilized life.*" [46].

As a result of this landscape history, it is now possible to distinguish in the region geometries of land, water and plantations, surrounded by a "minor" architecture made of rural houses, farms and villas of those who once were landowners, buildings which are in perfect respect of the landscape [48].

The reclamation territory of Veneto is therefore an interesting case study because it outlines a way of interpreting the landscape as a clear example of space where to see the relationship between natural, anthropic and cultural processes, "*where innovative approaches are strictly intertwined to new use practices by tourists, residents and metropolitan citizen.*" [46].

The research intention is to be able to understand the relationship that the inhabitants have with the landscape, how they interact with it, now and during the past. Moreover, the research will focus on the relationship between people and historic buildings located in the case study area and if actually living in a small city, with an important built historical context, is a determining factor for the psychophysical well-being of people, relating this concept to that of belonging to a place. In this regard, place identity has been considered as the way in which a place contributes to the identity of a person or people [20] and the composites of its characteristic features [49]. Research claims that identity in an urban environment is defined to a greater or lesser extent by the elements and activities of the environment or by events that take place within such an environment [50].

The impact that the identity of a place with a rich historical context has on man and not even the emotional sphere linked to the concept of partnership should not be underestimated; places play a vital role in developing and maintaining self-identity and the group identity of the people [51] and this is therefore a closely related factor in determining the psychophysical well-being of the citizen.

An additional aim was to recognize the current need for interpretation and representation of the dynamics and space of a city, of the relationship between built and natural, which drives us to combine a digital research method with a more traditional one, in order to develop a strategic visualization of these spaces capable of providing interpretations and possible advice for future developments. The documentation of landscapes through new technologies involves more and more integrated methodologies and graphic results such as Heritage-BIM, advanced 3D modeling and immersive environments. The process of mapping places is a necessary pre-condition for design action, according to recent UNESCO recommendations [52], and also includes the emotional and perceptive dimension, to represent space through visual thinking and produce graphic materials.

Through those emerging tools, cutting-edge practices can be inspired concerning not only agrarian and urban, but also historic urban landscapes, with the possibility of planning and preserving integrity and authenticity of heritages cultural landscape.

## 4. Materials and Methods

The research implementation started the application phase on the first case study, Piazzola sul Brenta, chosen for multiple reason: its valuable architecture, Villa Camerini-Contarini, around which the village historically revolves, green and blue spaces integrated with the built ones. All of these aspects will be taken into consideration and explored as regenerative elements, concentrating on the interaction between each of those elements and how this relation benefits the life quality of citizens.

In particular, the research also focuses on understanding whether it is actually more sustainable to live in a small village than in a large city, both from an environmental

and social point of view, evaluating the connection between people well-being and the surroundings. Concentrating on RT3 theme "Interaction between environments and human beings in the construction and sustainable design sectors", the first step was to cooperate with the Department of Psychology of Padua, which leaded to a systematic review of the literature on restorativeness. At the same time, historical research also began from a spatial point of view, taking into consideration the first case study, Piazzola sul Brenta. The characteristic of that small town in the Venetian countryside, which is built around Villa Contarini, offers a perfect example for this research.

It is possible to summarize into three phases the first set of the research process:

1. General considerations on the case study and systematic review of the literature regarding the history of Piazzola sul Brenta and restoration of green, blue and historical built spaces;
2. Overlapping historical maps: historical maps were used as palimpsest and compared with the state of the art.
3. Planning further steps of the research due to results of points 1 and 2.

Thereafter, the research will be divided by critically analyzing historical maps, state of the art and data acquired through surveys, and by spatially analyzing the historical built landscape and the natural landscape. Afterwards, the two case studies will be compared, in order to draw respective conclusions, based on the acquired data.

### 4.1. Case Study: Piazzola sul Brenta

The history of Piazzola sul Brenta overlaps with that of Villa Camerini-Contarini, one of the largest and most magnificent villas in the Veneto region, which it is assumed derives from the seventeenth-century transformation of a villa built by Andrea Palladio in 1540 for Paolo Contarini and his brothers [53]. Characteristic of the Villa is its outstanding and showy Baroque appearance and the vast park-garden of over 40 hectares with fishponds, lakes and tree-lined avenues. The town is also strongly related to surrounding natural elements: not far from the city center, the Brenta river, includes a natural park (Naturalistic Area of Piazzola sul Brenta) between its bends and on whose banks there is a long cycle path immersed in the greenery.

Between the seventeenth and eighteenth centuries, the Villa reached its maximum splendor; and in the seventeenth century the two side wings, the "barchesse" were added. With the works of the seventeenth century, the Villa was practically renovated also as regards the garden in front of the building, which was transformed into a single and large rectangular green space with a fountain in the center. The relationship that Villa Contarini proposes between public space and private space should also be underlined. In fact, the Villa opens onto a semicircular square, bordered by arcades under which there are still shops in the city market today.

Focusing on the relationship between the built environment and the people who live there, in the contexts of small villages and small towns, this is an experimental study on restorativeness and residential quality of the urban environment in Piazzola sul Brenta.

Starting the research process, the main purposes were listed to create a clear methodological process:

- To investigate the restorative effects of both the green and historical environment on psychological, physiological, and cognitive functioning of residents;
- To investigate whether the restorative effect of the green and historical environments change in social context, looking in particular at the humanistic concept of regeneration associated with gardens;
- To examine whether the restorative benefit of the park and the Villa is associated with residential satisfaction.

Regarding the latter point, variables typically related to residential satisfaction and place affection are central: spatial (i.e., architectural-planning space, organization and accessibility of space, green space), human (i.e., people and social relations), functional

(i.e., welfare, recreational, commercial, transport services), and contextual (i.e., pace of life, environmental health, upkeep) aspects. To achieve these goals, there is the need to examine the physical space in its historical and current dimensions, using historical documents and integrated survey, with the intention of understanding how built space and natural space dialogue with each other.

### 4.2. Historical Research

Keeping in mind this guideline, one of the first steps of the research was to analyze historical maps, to understand how the built and natural landscape had changed over time. The main elements on which the research focused in this first analysis are the bed of the Brenta river and Villa Camerini-Contarini. The research started studying historical maps retrieved in the State Archive of Venice, maps representing the Villa and the relationship between the Villa and the Brenta's river (Figure 1).

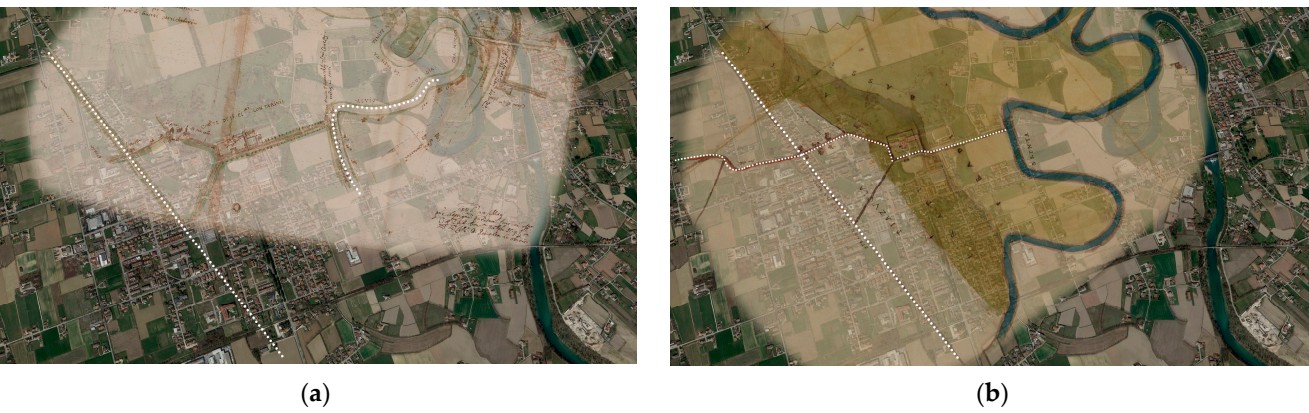

| (a) | (b) |

**Figure 1.** Historical maps as a palimpsest. (**a**) The old riverbed mapped in 1556, compared with the current one; (**b**) anthropic action on the natural element: the river diversion's plan after 1558.

From the observation of mainly two maps and their superimposition on the current state of the landscape, it can be seen how the bend of the Brenta river has changed its position and how a detour has been built, with the purpose to irrigate the fields. One of historical map used had been sent by Contarini family to the "Provveditori sopra Beni Inculti", which was a Judiciary founded in 1556 involved in the management of all inland waters and their exploitation for agricultural or industrial purposes (Figure 2).

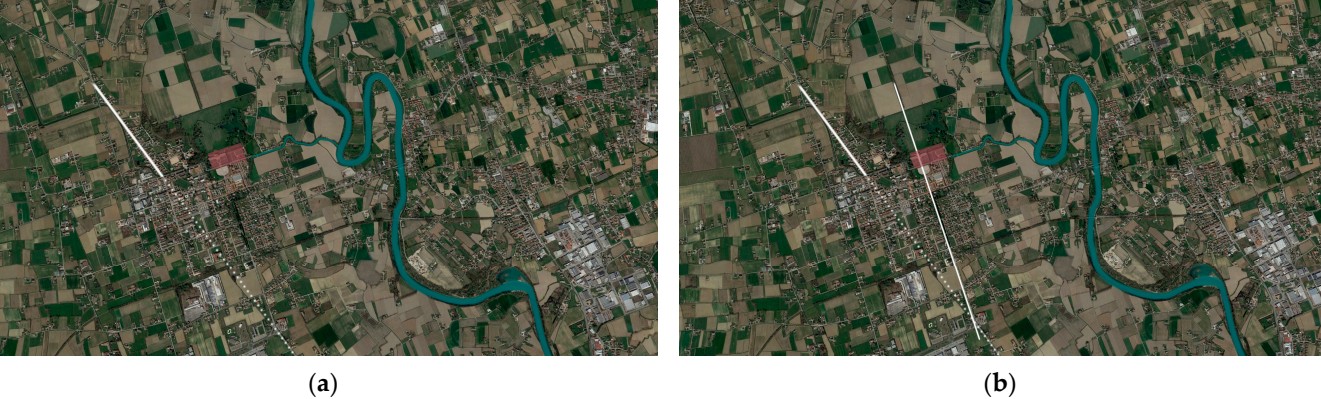

| (a) | (b) |

**Figure 2.** Overlapping of natural and historical landmarks. (**a**) Brenta's River and the Roman axis; (**b**) comparison between the orientation of Villa Contarini and the Roman axis.

Contarini's family followed the example of other patrician families who have been excavating on both sides since the fourteenth century of the course of the river, a series of irrigation ditches which draw water from the Brenta downstream of Bassano and the

distribute on their properties. The first excavation on the hydrographic right seems to have been the della Roggia Molina (1311) in the area of Grantorto and Cartura, followed by that of the Roggia Grimana Vecchia (1569), Roggia Grimana Nuova and Roggia Quadretti, more than a century late, just to mention the main ones; all watercourses that generally take their name from families that create or enhance them creating a small river, aimed at irrigating the fields north of the Villa. The presence of the Brenta river, so physically close to Piazzola, is an element that will be used also years later, for the jute factory; the river is therefore linked inevitably to the life of the citizens of Piazzola (Figure 3).

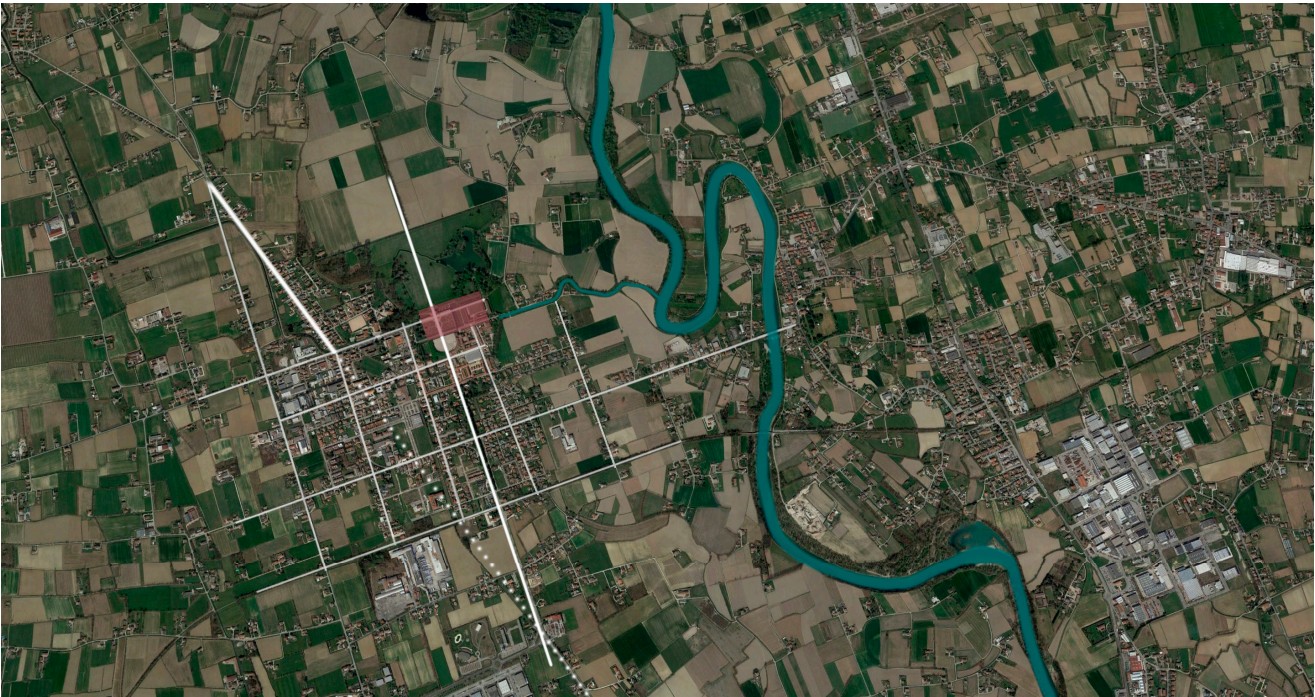

**Figure 3.** Overlapping of natural and historical landmarks. The urban grid of Piazzola is based on the orientation of the Villa.

Analyzing the anthropic space within the territorial framework of Piazzola sul Brenta, were identified the Roman axes passing through the village and then get lost. Additionally, the urban grid of Piazzola was identified and compared to the orientation of the roman axes, but they did not overlap. This initial analysis has led to an understanding of how historically relevant the presence of Villa Contarini is: in fact, the urban layout of the village has detached itself from the Roman axes and leans instead on the grid created by the orientation of the Villa.

The analysis of historical maps focused on understanding how natural landscape had changed and if it had changed due to human actions. The intention for future research is to shift the focus to the built landscape and analyze it to understand its dynamics.

## 5. First Results and Further Steps

As the first result of this ongoing research, the characteristic landmarks of the case study of Piazzola sul Brenta were identified. Using historical maps and analyzing them as a palimpsest, the history of the natural element that characterizes this small urban reality (the Brenta river) and the architectural element in which a large part of the identity of the place resides have been retraced. In fact, this analysis has revealed how strong the presence of Villa Contarini is for Piazzola, since the entire urban area has a grid based on the orientation of the Villa itself, detaching itself from the ancient axes of the Roman age. By developing the research described above, it appeared also how little interest has so far been focused on the topic of the possible regenerative nature of staying in an environment

of historical value and architectural value. The focus usually shifts to the regenerative value of the natural, green and blue environment, or even to the newly built environment, leaving out the discussion of the existing historical built environment.

These are only the initial results of this ongoing research, which has just begun, which is developed in the context of an inter-university project and which is characterized by a strong interdisciplinarity.

Future development will be focused on the digitization of case study' landscape, at a urban scale, in order to create a set of 3D models and integrated digital data to be combined with historical sources. The creation of parametric models will enable the aggregation of different information to geometric data, with the possibility of better understanding the dynamics between man/built/nature in every single case studies and, by comparing these data, to draw the necessary conclusions aimed at a better and sustainable planning for the future.

As further steps, the research intent to be carried on proposes also an application level, in cooperation with the Department of Psychology of Padua (DPG), which will first focus on a study of images in an immersive room (photographs of greenery and buildings) to understand the participants' reaction to them; subsequently the DPG will lead to an experiment on site, using electrocardiogram devices (ECG) to measure the physiological index of the participants during the visit of the Villa and the park.

The Department of Civil, Construction and Environmental Engineering of Padova (DICEA) and the Department of Cultural Heritage of Padova (DBC) will instead address the survey and representation, both of the landscape and of the historic buildings. The procedure to be followed can be summarized in three points, the first of which has already been partially carried out, using and studying the historical maps and archival documents, followed by the second, data acquisition, and by the third and last data integration and modelling.

Regarding how natural and artificial elements had transformed through time, three questions in particular will lead the research:

- How is the artificial element inserted in relation to the natural one?
- Where is the crossing point of the two spaces?
- What is the regeneration value of the artificial/architectural element?

In the further steps the aim is to use digital and traditional research to carry out an analysis of art, architecture and urban history. This field welcomes submissions from a range of disciplines fundamental to this topic of the digital and material: History of art, History of architecture and of the city, Representation (in particular, architectural surveying, Building Information Modeling—BIM, Geographic Information System—GIS, Perspectival and Photographic restitutions).

The next step of the research will be to analyze the space first of Piazzola del Brenta, then of Aquileia. Using the integrated survey methodology, both natural and built spaces will be surveyed, focusing on the buildings of historical and artistic value on which the research focuses. Then, using BIM modeling, the research will lead to the use a 3D digital reconstruction of historic building, based on the data acquired in the survey phase.

The research will be divided by spatially analyzing the historical built landscape and the natural landscape, critically analyzing historical maps, state of the art and data acquired through surveys. Subsequently, by comparing the two case studies, we intend to draw respective conclusions, based on the acquired data and on this comparison. To have precise results of this comparison, we intend to use the 3D model that will be made using BIM technology, for an accurate analysis that will combine multiple data of a different nature: spatial, social, interaction between the environment and manmade elements. In fact, it is precisely the character of interoperability that is central to this research, which seeks to carry out a comprehensive analysis with the aim of understanding the possible benefit that contact with a built environment of historic value can have on who interacts with it.

Developing the concept that digital and material is of current relevance to the field of art and architectural history, three different supports to achieve the final goal will be used (Figure 4):

- Data retrieval through archival research;
- Data acquisition through laser scanner technologies and photogrammetric surveys;
- Data processing and creation of 3D integrated models;
- Data management and models' assessment through interoperable platforms;
- Integration of data as a means of analysis in the conservation process of architectural heritage with the virtual reconstruction of architectural features.

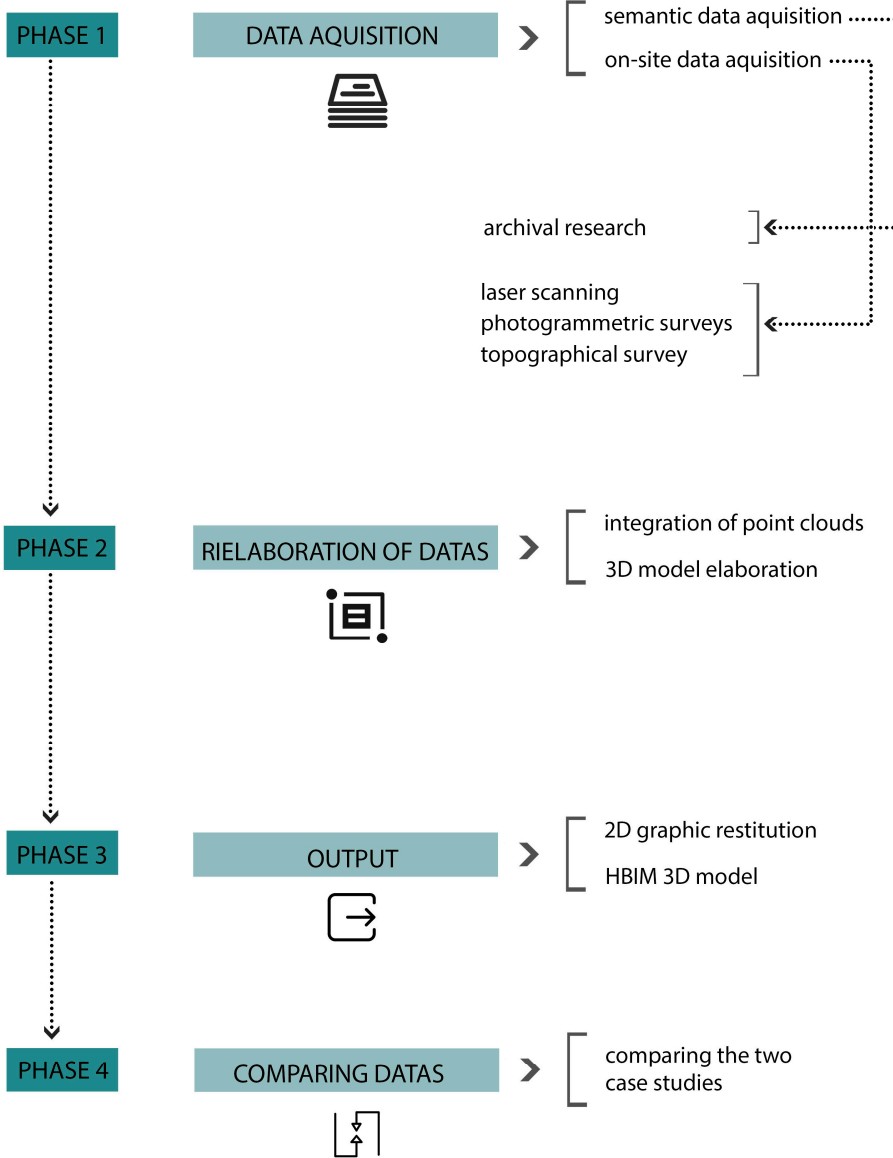

**Figure 4.** Workflow: phased research development program.

## 6. Discussion

The multi-disciplinary scope of this research opens up a broad discussion, which extends to different topics. The breadth of topic of RT3.1 stimulated to reflect on the extremely vast theme of the interaction between environments and human beings in the construction and sustainable design sectors, from which the research then focused on the topic of the restoration of places that present a union of natural and historical valuable architectural elements. Coordinating with more researchers from different disciplines is not an easy task but gives the possibility to learn and to think in a transversal way, looking

at the problem to be solved in its various aspects. Improving the quality of life for the future is for sure a need of the present, knowing that the trend of urbanization, metropolis living, and climate change is related to psychologically stressful situations as a result of environmental stressors. Dealing with those stressors calls for proof from numerous fields, which includes psychology, architecture, design, and information technology [17].

The aim to investigate the restorative effects of both the green and historical environment on psychological, physiological, and cognitive functioning, could be a very useful analysis to understand the places, the relationship that we have with them and thus to the future design and safeguard of these same places. Understanding whether the restorative effect of the green and historical environments change the social context, examining whether the restorative benefit of nature and historical architecture is associated with residential satisfaction, are goals meaningful not only to evaluate the state of the art of the case studies, but also valuable to create a methodology that can be used in the field of future safeguards. Many are the variables typically related to residential satisfaction and place attachment: spatial (i.e., architectural-planning space, organization and accessibility of space, green space); human (i.e., people and social relations), functional (i.e., welfare, recreational, commercial, transport services), contextual (i.e., pace of life, environmental health, upkeep) aspects. To connect all those different data there is the need of support from technologies that can process different elements, in order to outline guidelines and recommendations for the improvement of green and built areas for the future municipalities.

In this field a topic of great interest is also to understand how augmented reality (AR) and virtual reality (VR) also interacts with man and nature. The creation of a virtual space affects the topic of perception of the physical space, its use, its feeling. The difference between VR and AR in terms of the possibility of relating directly to the heritage environment being enjoyed, and the need to do so in situ in the case of AR, is of vital importance [54]. However, as discussed in the essay *"Reconnecting the city: the historic urban landscape approach and the future of urban heritage"* [54], VR and AR can have a great influence on the perception of space and not always a positive one, often due to the mere "virtualization" of space [54], with the risk of becoming alienated from the real context. On the contrary, however, this research wants to understand what is the key to the phyco-physical well-being of a citizen who lives in a small historic urban context, in relation to the natural environment, using new technologies as support of critical reading of the data.

## 7. Conclusions

A new research method on the topic of cultural heritage will be investigated and experimented, referring to built and natural spaces, located in small the urban context, considering primary sources—archival documents, images, photographs—stored in digital databases as the fundamental basis for historical research regarding architectural heritage [55] (pp. 184–200). The current need of representation of landscape puts us in front of the problem of the digitalization of data: as it is emphasized in Amoruso and Salerno's book [56] *"digitization doesn't consist in fact in a trivial "translation" from analogic to digital in order to get digital copies of text, drawings, maps, videos.".* New research frontiers require finding new modalities of representation, compatible with the analyses carried out, accessible and interoperable.

Furthermore, 3D digital reconstruction of historic buildings with reference to these image and data databases aims to find a new approach capable of an interactive contextualization and intuitive access to often-inaccessible data. Therefore, this method merges scientific research and the study of historical sites, basing the research on advanced technological opportunities such as use of interoperable and parametric modeling (specifically Building Information Modeling—BIM), exploiting its ability of the interconnection of different types of data. Creating digital models means to create a tool that can be used not only for historic architecture's representation and documentation [57] (pp. 433–438), or to create databases of high-definition, three-dimensional morphometric data [58] (pp. 225–244) but

also as a tool to experiment and to merge data of different nature, able to compare more case studies in order to have a thorough research.

An additional contemporary need is to clarify the restorative effects of different urban environments: green spaces, blue spaces, and historical places on individuals. Previous studies have focused only on one of these outcomes (e.g., mental health) or on one or two types of environment (e.g., green and urban) and some reviews or meta-analysis have compared the restorative effects of natural and built environments [28,46] or have focused only on the impact on mental well-being or cognitive function [6,59,60]. The purpose is to systematize whether and how green, blue and historical environments can positively influence individual and social functioning from a physiological, cognitive, emotional and relational point of view, in order to outline guidelines and recommendations for the improvement of green and built areas for the future municipality.

The desire to direct research in this field comes from the awareness of a lack of knowledge regarding the restoration effects of places where the natural space interfaces with the historical valuable architecture and the belief that a study and analysis of this kind could be fundamental for the future planning in a careful, sustainable way, capable of safeguarding the authenticity of historical places and landscape.

**Author Contributions:** Conceptualization, G.M.; methodology, G.M. and A.G.; software, G.M.; validation, F.M., A.G., G.G. and E.S.; formal analysis, G.M., A.G., G.G. and E.S.; investigation, G.M., A.G., G.G. and E.S.; resources, G.M., F.M., A.G., G.G. and E.S.; data curation, G.M.; writing—original draft preparation, G.M.; writing—review and editing, F.M.; visualization, G.M.; supervision, A.G., F.M., G.G. and E.S.; project administration, G.M. and A.G.; funding acquisition, G.M. All authors have read and agreed to the published version of the manuscript.

**Funding:** This research was funded by PNRR, Missione 4: Istruzione e ricercar; Component 2: dalla ricerca all'impresa; Investment n° 1.5 funded by the European Union—NextGenerationEU. Progetto iNEST (Interconnected Nord-Est Innovation Ecosystem); ecosystem (Area tematica: EI).

**Institutional Review Board Statement:** Not applicable.

**Informed Consent Statement:** Not applicable.

**Data Availability Statement:** Final data of the research will be published on the website of iN-EST's project, in particular regarding the topic of Spooke 4: https://www.consorzioinest.it/en/city-architecture-and-sustainable-design-2/.

**Acknowledgments:** This research is being developed within the Ph.D. research of Greta Montanari, with Supervisor Andrea Giordano and Co-supervisor Federica Maietti. The host Department is ICEA, at the University of Padua. This research is conducted in collaboration with two History Professors: Gianmario Guidarelli, ICEA Department of Padova, and Elena Svalduz, BC Department of Padova. The scholarship belongs to the 38th PhD Cycle, it is entitled *"Comprendere le complessità del Patrimonio Culturale: visualizzazione digitale e comunicazione interoperabile di architetture storiche nei sistemi urbani"*; it is financed by PNRR funds, inside the iNEST research project (INTERCONNECTED NORD-EST INNOVATION); ECOSYSTEM (Thematic area: EI). Within the INEST (Interconnected Nord-Est Innovation Ecosystem) project, this research is located within Spoke 4—City, Architecture and Sustainable Design, in particular it refers to RT3.1—Interaction between environments and human beings in the construction and sustainable design sectors. All partners can be found at the link of iNEST project, in the References section.

**Conflicts of Interest:** The authors declare no conflict of interest.

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
