# Peer review of "A Strategic Interpretation of Landscape through Interaction between Natural, Built and Virtual Environments: The Case Study of Piazzola sul Brenta"

_sustainability, doi:10.3390/su151813445_

Round 1

Reviewer 1 Report

Dear authors,

After close reading I have to agree that the project you are a part of is authentic and that all disciplines sharing holistic view on engineering will prospect with  ideas and goals that your project holds and carry.

Thus, the article is valuable asset and needs to be presented in all its parts. Unfortunately, it is not coherent and is very hard to read. It is impossible to divide the overall idea of the project apart from this particular stage and investigation inside of it. Therefore, albeit valuable it needs refinement and thoroughly restructuring with additional results as to hold the clarity and multi-disciplinarian unity.

Title - a strategic interpretation of landscape through interaction between natural, built and virtual environments: case study…… ( one needs to see that it is about digital data analysis, that it is multidisciplinary and that also run through case study )

Abstract - problem and intention, goal and method, results. Also pinpoint that it is a part of the research project also mention which part and how log it lasts

Structure and content - part 1 and 2 are ok. Part 3 should be much longer and in detail observing on how the collaboration function. It is a place to explain all tree environments and their roles in the context of partnership and their interaction. Lines 211 to 215 needs to be elaborated with additional resources and reference literature. Also 217 to 221 extend with additional explanation on how and why and through what … discussions with partners, seminars….. Part 4 have to be in line with the promised ( in abstract and intro part) and has to explain the process in detail on an abstract level and then to show the procedure on the case study again in more detail and with designed graphic illustrations. Line 302 to 314 represent a unique discussion and therefore deserve a whole subtitle rather then a paragraph. Part 5 could be incorporated with part 4. Part 6 is out of reach as it goes way beyond the scope of the article.
In my opinion, the article should focus on the analysis done multidisciplinary and positioned as a part of the research project. In present state, it looks more as a mid term project report with possible future perspectives. It is important to show the process of multidisciplinary analysis as detailed as possible.

Reviewer 2 Report

The item of the paper is interesting, however the paper, for having the possibility to be considered for the publication has to be completely rewritten.

In the present form it looks like the illustration of the research framework of a project iNEST project (Interconnected Nord-Est Innovation Ecosystem), financially supported in the frame of PNRR Program, more than the description of the results and discussion obtained in the frame of a deliverable of the project, with a complete meaning by itself.

The Authors wrote that the aim of the research is to analyze what role the union of the natural element and the valuable artificial element ( i.e. historical valuable architecture, art ) could have in generating health and psycho-physical well-being, but this does not appear in the paper.

The project is clearly at the beginning and the dissemination actions, as well known,  are necessary to the prosecution of the project, however they have to be performed in a coherent way.

The title has to be redesigned being more attinent to the content of the text.

The abstract has to evidence better the content of the research and evidence the part that deals with the exposed work and the fact that the results are preliminary.

Due to the experimental findings illustrated below, the section 3. An interaction between natural, built and virtual environments, should be extended.

The Discussion is not a coherent discussion of the case study but an illustration of another part of the project and can be omitted or has to be totally modified.

The conclusions are missing.

Reviewer 3 Report

The following items have been sent to improve the structure of the article:

1- Make the title of the article more readable. (The title of the article is incomprehensible)

2- The necessity and importance of the issue should be clearly stated. In the introduction section

3- It is better to mention the research questions clearly in the introduction section.

4- The article has not yet covered the background of the research. Be sure to add the research background to the article.

5- In the research method section, the statistical population, the scope of the research, and the data analysis method should be explained exactly.

6- It seems that scientific analysis is still not seen in the data analysis section. It is necessary to analyze the data based on a scientific method. With diagrams and pictures

Good luck

 Minor editing of English language required

Round 2

Reviewer 1 Report

Dear Authors,

Thank you for the letter. The article has been improved gradually. It is very well written and focused. Very best in the next fazes of research project.

Reviewer 3 Report

The changes made in the article are approved

 Minor editing of English language required